# Leveraging Affinity Interactions to Prolong Drug Delivery of Protein Therapeutics

**DOI:** 10.3390/pharmaceutics14051088

**Published:** 2022-05-19

**Authors:** Alan B. Dogan, Katherine E. Dabkowski, Horst A. von Recum

**Affiliations:** Department of Biomedical Engineering, Case Western Reserve University, Cleveland, OH 44106, USA; abd51@case.edu (A.B.D.); ked87@case.edu (K.E.D.)

**Keywords:** antibody, affinity, cyclodextrin, protein therapeutics, sustained drug delivery

## Abstract

While peptide and protein therapeutics have made tremendous advances in clinical treatments over the past few decades, they have been largely hindered by their ability to be effectively delivered to patients. While bolus parenteral injections have become standard clinical practice, they are insufficient to treat diseases that require sustained, local release of therapeutics. Cyclodextrin-based polymers (pCD) have been utilized as a platform to extend the local delivery of small-molecule hydrophobic drugs by leveraging hydrophobic-driven thermodynamic interactions between pCD and payload to extend its release, which has seen success both in vitro and in vivo. Herein, we proposed the novel synthesis of protein–polymer conjugates that are capped with a “high affinity” adamantane. Using bovine serum albumin as a model protein, and anti-interleukin 10 monoclonal antibodies as a functional example, we outline the synthesis of novel protein–polymer conjugates that, when coupled with cyclodextrin delivery platforms, can maintain a sustained release of up to 65 days without largely sacrificing protein structure/function which has significant clinical applications in local antibody-based treatments for immune diseases, cancers, and diabetes.

## 1. Introduction

Protein and peptide therapeutics have emerged in the past few decades as a major branch of modern pharmacology. Due to their modularity, high specificity, and low off-target activity, protein therapeutics have been utilized to treat diseases including cancers, genetic disorders, autoimmune diseases, and inflammatory diseases by replacing, augmenting, or interfering with cellular pathways or by delivering payloads to achieve a desired outcome. In 2019, protein-based therapeutics contributed to eight out of the top ten drugs by sales globally and encompasses an annual market of around 69.4 billion dollars [1]. A majority of these novel therapeutics, consisting of around 100 FDA-approved therapeutics since their introduction in the 1980s, leverages the high target specificity of monoclonal antibodies (mAbs) and antibody fragments (FABs) [2]. Monoclonal antibody therapeutics have been shown to efficaciously treat autoimmune diseases, cancers, asthma, and have even shown significant potential for treating COVID-19 [3,4,5].

Despite these advantages, many protein therapeutics suffer from relatively short in vivo half-lives compared with disease prognosis and high accumulation in the liver and spleen [6,7]. This translates to clinical protocols that typically involve parenteral drug administration; however, frequent injections decrease patient compliance, increase risk of infection, and promote nonspecific drug targeting—all of which greatly decrease therapeutic efficacy. Therapeutic efficacy of protein-based drugs with bolus administration is also highly dose dependent, which can also decrease positive patient outcomes. Therefore, there is a need for extending and controlling the delivery of protein and peptide therapeutics.

Polymeric drug delivery systems present a possible solution to extending the delivery of protein and peptide therapeutics. Hydrogels and in situ forming implants, which are composed of polymer matrices such as poly(lactic-co-glycolic acid) (PLGA), have been used to locally deliver small-molecule and protein payloads via diffusion-based mechanisms [8,9]. Nanoparticles and liposomes have also been used to deliver proteins by taking advantage of the enhanced permeation and retention effect and can greatly increase in vivo circulation time [10,11]. However, the mechanism of delivery for a majority of these solutions involves diffusion, degradation, and/or encapsulation, which makes existing protein delivery systems effective for bolus-injection therapeutics but unattractive for sustained, controlled, local delivery.

Affinity-based drug delivery systems have emerged as a popular way to help control and prolong the release of protein therapeutics [12]. Broadly, “affinity-based” delivery encompasses any matrix/guest system that leverages receptor/ligand, electrostatic/ionic, hydrogen bonding, hydrophobic, or van der Waal interactions to increase the thermodynamic interactions between the matrix host and the guest payload [13,14,15]. A well-established example of this is synthesizing a polymer matrix with exposed streptavidin sites designed to bind with biotinylated protein payloads [16,17]. However, due to the strong interactions between biotin and streptavidin (dissociation constant [K_D_] ~ 10^−15^ M), much of the protein payload remains associated with the streptavidin matrix and is never released where it can interact with cells and tissue [17,18].

Polymerized cyclodextrin (pCD) is an alternative affinity-based drug delivery system that utilizes hydrophobic interactions between cyclodextrin’s hydrophobic “interior” pocket and hydrophobic payloads [19,20,21]. Typically, this interaction has been used to enhance loading and extend the release of small-molecule, hydrophobic drugs, as the inclusion pocket of cyclodextrin only ranges from 0.5–0.8 nm in diameter [22]. pCD has been shown to have drug loading and drug release kinetics directly associated with the K_D_ of its included payload and has the ability to refill in vivo, making it especially useful for long-term therapeutic timelines [23,24,25,26,27,28,29]. While many groups have investigated pCD to deliver small molecules such as rifampicin and other smaller therapeutics such as RNA, the delivery of proteins by pCD, to the best of our knowledge, has never been achieved due to their large molecular weights and hydrophilic nature of the therapeutics [30,31].

Recently, our group has shown that conjugating polymer chains capped with “high affinity” groups can increase the overall conjugate’s affinity towards pCD polymers [32,33]. Briefly, by taking advantage of the strong interactions between adamantane (Ad) hydrocarbons and beta-cyclodextrin (K_D_ ~ 5 × 10^−4^ M), we have shown that the affinity between pCD and the drugs rifampicin and rapamycin can be increased by conjugating polymer tethers end-capped with adamantane groups (referred to as polymer-Ad). The drug–Ad conjugates exhibited a higher loading capacity and an extended window of release in pCD hydrogels [32,33], thereby suggesting that by manipulating the K_D_ between the payload and pCD matrix, we are able to modulate delivery and loading.

As protein and peptide therapeutics now constitute the current, and possibly future, direction of pharmaceuticals, we proposed to apply the idea of modulating affinity to hydrophilic payloads, such as proteins, so that they can take advantage of the “affinity-driven” loading and release kinetics common to pCD hydrogels. Many current protein delivery systems, such as PLGA hydrogels, are single use and may degrade into unwanted byproducts, many of which can negatively impact the overall therapeutic efficiency of the payload [34]. pCD hydrogels, when prepared with a non-cleavable crosslinker, do not degrade and can be reused and refilled with drugs after initial administration by nearby bolus injection, making it ideal for long-term therapeutic treatments [35,36].

Herein, we use bovine serum albumin (BSA) as a model protein, and monoclonal antibodies (mAb) as a functional example to show that proteins can be delivered from pCD hydrogels by increasing the protein’s affinity for the system through the conjugation of polymer–Ad groups (Figure 1). Additionally, we explore the impact that conjugating polymer–Ad groups have on the functionality of mAbs to help assess if pCD can be a viable platform for the clinical delivery of therapeutic proteins and peptides.

## 2. Materials and Methods

### 2.1. Materials

Bovine Serum Albumin (BSA) heat shock fraction, protease free, fatty acid free, essentially globulin free, pH 7, >98% and QuantiPro BCA Assay Kit were purchased from Sigma-Aldrich (St. Louis, MO, USA). IL-10 ELISA Kit (Interleukin 10) was purchased from Antibodies-online.com accessed on 12 February 2020 (Limerick, PA, USA; MabTag). *β*-cyclodextrin (*β*-CD) prepolymer, lightly crosslinked with epichlorohydrin, was purchased from Cyclolab (Budapest, Hungary). Mini-PROTEAN TGX Precast Protein Gels (4–20%) and Precision Plus Protein Dual Color Standards were purchased from Bio-Rad (Hercules, CA, USA). Maleimide PEG Hydroxyl (MW = 5000 g/mol) was ordered from JenKem Technology USA (Plano, TX, USA). All other reagents, solvents, and chemicals were purchased from Fisher Scientific (Hampton, NH, USA) in the highest grade available.

### 2.2. Protein–PEG–Adamantane Conjugation Synthesis

To synthesize adamantane-capped PEG (Ad–PEG_5000_–Mal), a 1:1 (*w*/*w*) ratio of adamantane carbonyl chloride and maleimide–PEG_5000_–hydroxyl was dissolved in anhydrous chloroform (5 mL chloroform per 1 g mixture), according to a previous protocol [28,37]. Equal molar amounts of triethylamine to PEG were then added dropwise and the solution was allowed to react for 24 h at room temperature (RT) under agitation (Figure 2a). Once reacted, the product was precipitated with diethyl ether three times (using half of the total volume of chloroform) and solvent was removed in vacuum at 25 °C overnight (ON). As pCD is a covalently bonded polymer network with closely-packed monomer units, PEG_5000_ was selected for adamantane conjugation as its estimated hydrodynamic radius (~2 nm) was predicted to sufficiently “space out” the adamantane groups from the hydrophobic interior of folded proteins (3–14 nm). This was carried out to help ensure adamantane groups were available for complexation within cyclodextrin’s interior.

Protein–PEG_5000_–Ad conjugates were synthesized with sulfhydryl-reactive crosslinker chemistry. Protein samples (Bovine Serum Albumin, BSA, or IL-10 monoclonal antibody, mAb) were incubated with 10x molar excess TCEP (tris(2-carboxyethyl)phosphine) slurry for 1 h at RT on an end-over-end mixer. Once disulfide bonds were reduced, a variable molar excess (2×, 4×, 8×) of Ad–PEG_5000_–Mal was added to the protein samples and were allowed to react for 4 h at RT (or ON at 4 °C) (Figure 2b). The protein–polymer conjugates were then dialyzed for 48 h (MWCO = 14 kD). In this study, BSA–PEG_5000_–Ad and mAb–PEG_5000_–Ad were investigated. The concentration of mAb present in each reaction was according to MabTag kit standards.

### 2.3. Nuclear Magnetic Resonance

Nuclear magnetic resonance (NMR) was used to verify successful conjugation during synthesis steps. All spectra of presented chemical species were recorded by Bruker 300 MHz NMR system (Bruker, Germany) in DMSO-*d*_6_ or D_2_O solvent, as indicated.

### 2.4. Rate of Hydrolysis for (Ad–PEG_5000_–Mal) Species

An amount of 10 mg of Adamantane–PEG_5000_–Mal was dissolved in 10 mL of D_2_O and incubated to 37 °C, agitated at 100 rmp. An amount of 600 μL aliquots were sampled at designated intervals and ^1^H-NMR spectra were obtained. Percent hydrolysis was determined by comparing the integral of the “hydrolysis alcohol peak” at δ = 6.1 ppm and a fixed peak at δ = 4.2 ppm. Samples were capped with N_2_ after obtaining each sample.

### 2.5. pCD Polymer Synthesis (MPs and SRPL)

pCD polymer microparticles (MPs) and Sustained Release Polymeric Liquid (SRPL), a fluidic pCD formulation, were synthesized to serve as insoluble drug delivery vehicles [38]. As we aimed to investigate the impact of polymer topology on drug loading capabilities for protein payloads, SRPL was used as an example of crosslinked, linear branch topology while MPs were used to represent crosslinked, spherical topology. As polymer macrostructures are widely known to influence drug loading and drug release profiles, we aimed to investigate the differences between a linear, viscous polymer and a compact, dense particle, respectively.

Briefly, microparticles were synthesized from epichlorohydrin-crosslinked *β*-cyclodextrin prepolymer solubilized in 0.2 M potassium hydroxide (25% *w*/*v*) preheated in a 60 °C oil bath for 10 min. Light mineral oil was then added to a beaker with Tween 85/Span 85 (24/76%) and stirred at 500 rpm. Ethylene glycol diglycidyl ether (0.01 M) was added to the prepolymer solution dropwise and then mixed. After vortexing, the prepolymer solution was added to the oil/Span/Tween 85 solution and heated in a 70 °C-oil bath. The stir speed was increased and then the mixture was stirred for 3 h. After incubating, the particles were taken out and centrifuged at 200× *g* to be separated from the oil mixture and then washed with excess hexanes twice, excess acetone twice, and finally diH_2_O twice. The microparticles were then frozen and lyophilized before further use. Based on previous studies, we can estimate that this protocol generates pCD microparticles with a size of 81.88 ± 36.86 μm with a polydispersity index of 0.2 [39].

SRPL was synthesized with *β*-cyclodextrin crosslinked with epichlorohydrin, dried for 4 h at 70 °C, and then stirred (150 rpm) in a 55 °C-oil bath for 10 min. DMSO (4 mL per gram of dried CD) was then added to the CD and incubated for 10 min. Hexamethylene diisocyanate (HDI) crosslinker (45 μL HDI per gram dried CD) was added to the CD mixture and capped with N_2_ gas. The speed of mixing was then increased and after two 15-min intervals, the vial was checked to observe increases in viscosity until the solution appears viscous and glassy (~30 min). Then, diH_2_O was added to quench the remaining crosslinker and synthesized polymer was lyophilized overnight (ON).

### 2.6. Affinity Testing—Surface Plasmon Resonance

Experimental “affinity” between *β*-CD monomers and protein conjugates were measured experimentally through surface plasmon resonance (SPR) with a Biacore X100 system (GE Healthcare Bio-Sciences, Pittsburgh, PA, USA) according to previous protocols [40,41]. The surface of a sensor chip CM-5 was conjugated with EDC (0.4 M) and NHS (0.1 M) followed by 10 mM of 6-amino-6-deoxy-*β*-cyclodextrin (CycloLab, Budapest, Hungary) suspended in HBS-N buffer (a HEPES balanced salt solution with pH 7.4). The other channel was conjugated similarly with aminodextran (Thermo Fisher Scientific, Waltham, MA, USA) to determine specific versus nonspecific interactions with a chemically similar but non-affinity substrate. The remaining functional groups were capped with ethanolamine. A multi-cycle kinetic experiment was performed with drug dissolved in a MilliQ water solution and was regenerated with 100 mM sodium hydroxide between samples. The differential responses between the channels were fit to steady-state affinity using Biacore evaluation software. Indicated K_D_ values (*) were within model confidence interval (Chi^2^ values below 10% of the maximum SPR response) [20]. A concentration range of 0.125–10 nM was used.

### 2.7. Drug Loading and Release

Affinity and drug loading and release kinetics of protein–(PEG_5000_–Ad)_x_ was tested in vitro in both pCD MPs and SRPL. BSA was used as a model protein, as it is relatively inexpensive and easy to detect with BCA assay kits. An amount of 10 mg of dried pCD MPs and SRPL, respectively, were soaked for 48 h in drug solutions (100 μg/mL) of each protein-conjugate species. The loaded pCDs were then washed 3 times with 1 mL of 0.1 sodium phosphate buffer and then incubated in 1 mL of a “physiological release buffer” (phosphate-buffered saline (PBS) and 0.1% Tween^8^°) at 37 °C on a rotary shaker. To simulate “infinite sink” in vivo conditions, aliquots were sampled frequently at recorded time intervals. BSA-conjugate concentrations were measured using a Micro BCA kit that gave a colorimetric readout that was compared with a standard curve for each BSA-conjugate species. Affinity interactions were “maximized” by keeping the number of potential “affinity-groups” under the theoretical maximum “host-groups” (e.g., BSA–(PEG_5000_–Ad)_2_ has two potential binding sites). Calculations were based on the following assumptions: *β*-CD MW = 1135 g/mol, Epichlorohydrin MW = 92 g/mol, BSA MW= 65,000 g/mol, and Mal–PEG_5000_–Ad at 5330 g/mol.

After the drug release aliquots decreased below detectable concentrations, the polymers were incubated for 96 h in 1 mL of DMSO to extract remaining protein conjugates. DMSO was removed from samples using a SpeedVac concentrator, ON, and reconstituted in 200 μL PBS. “Total loading” values were obtained by combining cumulative drug release results with DMSO-released drug.

### 2.8. Modified ELISA

To quantify antibody functionality after conjugation of affinity groups, we utilized a modified ELISA protocol to investigate mAb antigen recognition after PEGylation. Briefly, coating-capture antibody and blocking steps were performed according to the MabTag ELISA protocol and 500 pg/mL of rhIL-10 standard was added to each well. Detection-antibody was treated for 1 h with 10× molar excess TCEP and incubated for 4 h at RT with varying molar excess of Mal–PEG–Ad. Upon reaction completion, 100 μL of modified detection-antibody was added to each well, and each group (*n* = 3) was normalized to unmodified detection antibody. “Antibody functionality” was reported as a percentage of positive controls, which were assumed to have 100% antigen recognition.

### 2.9. Statistical Analysis

All statistics were calculated in Origin (OriginLab, Northampton, MA, USA). Statistically significance was defined as *p* < 0.05 with further specifications stated in figure captions. Drug release curves were analyzed by one-way ANOVA and Tukey post hoc test (*p* < 0.05).

## 3. Results

### 3.1. PEG–Adamantane (PEG–Ad) “Tether” Synthesis

Synthesis of Ad–PEG_5000_–Mal was confirmed via ^1^H NMR. Confirmation of Ad conjugation to maleimide–PEG_5000_–hydroxyl was observed as the coexistence of adamantane’s hydrocarbon (-CH_2_) peaks at δ = 1.7–1.9 ppm and PEG’s repeating backbone (-CH_2_-) peak at δ = 3.6 ppm. Reaction yield was determined by comparing the peak integrals of adamantane (δ = 1.7–1.9 ppm; 16 hydrogens) and the last carbon of PEG (δ = 4.2 ppm; 2 hydrogens), and a range of 73–96% conversion (*n* = 2 batches) was observed (Figure 3).

### 3.2. PEG–Ad “Tethers” Remain Stable for up to 40 Days

As Ad–PEG_5000_–Mal contains an ester group prone to hydrolysis, the stability of the chemical species was observed over time via ^1^H NMR in D_2_O. The integral of the terminal hydroxyl peak (-OH) at δ = 6.1 ppm was compared with a constant PEG (-CH_2_-) peak at δ = 4.2 ppm, and a percent of species hydrolysis was obtained. Over time, the integral of the terminal hydroxyl peak increased until it equaled half of the integral of the δ = 4.2 ppm, indicating 100% species hydrolysis. We found that at around 40 days, the adamantane is completely dissociated from the PEG linker, which is most likely attributed to the partial solubility of Ad–PEG_5000_–Mal (Figure 4). Hydrolysis was confirmed to have zero-order kinetics, as the percent of species exhibited a linear relationship relative to time.

### 3.3. PEG–Ad “Tethers” Successfully Conjugated to Reduced Proteins

To ensure maleimide–sulfhydryl chemistry was successfully taking place, nonreducing SDS-PAGE gels were used to quantify the molecular weight of our BSA and mAb protein–polymer conjugates (Figure 5). An increase in molecular weight was observed for all protein–polymer species, and the estimated conjugation ratio (*ECR*) of each species was obtained by the following equation:(1)$$ECR=\frac{MW_{conjugate}-MW_{protein}}{MW_{polymer\;’tether’}}$$

### 3.4. PEG–Ad “Tethers” Greatly Increased Protein–Polymer Conjugate Affinity for β-CD

Affinity between our protein–polymer conjugates and *β*-CD was tested using surface plasmon resonance (SPR). Unmodified protein species had characteristically low affinity for *β*-CD, while an increase in Ad–PEG_5000_ conjugation corresponded to an increased affinity (decreased K_D_). However, we found that as the number of “affinity” groups increased, we were unable to obtain low Chi^2^ value curve fittings, as indicated in Table 1. K_D_ values were determined from steady-state affinity equation curve fitting according to standards set in Biacore’s evaluation software.

### 3.5. PEG–Ad Conjugation to BSA Greatly Increases Loading Capacity into pCD, Regardless of Form

BSA–(PEG_5000_–Ad)_2_ was observed to have superior loading over both BSA–(PEG_5000_–Ad)_4_ and BSA–(PEG_5000_–Ad)_8_ (Figure 6). As BSA–(PEG_5000_–Ad)_4/8_ had comparable loading, we predict that both species were able to maximize the number of inclusion complexes formed with pCD hydrogels; however, the increase in PEG–Ad groups sterically hindered their ability to diffuse deeply into pCD. While MPs were observed to have slightly higher loading of BSA and BSA–(PEG_5000_–Ad)_2_, there were no significant differences in loading between SRPL and MPs in BSA–(PEG_5000_–Ad)_4/8_.

### 3.6. Protein–Polymer Conjugates Can Be Delivered for up to 65 Days in pCD Polymers

We observed that all BSA conjugates released for a significantly longer time (45–65 days) than BSA controls (1–2 days). While all groups experienced a “burst” release within days 1–4, a prolonged “affinity” release was observed after day 6 (Figure 7b,d). BSA, which lacks the ability to form inclusion complexes with pCD, was observed to only exhibit a brief, diffusion-based release. However, the extended release observed by BSA conjugates were most likely the combination of diffusion, “affinity” from adamantane groups complexing with pCD, and subsequent hydrolysis of adamantane groups. While MP pCD was found to have more favorable loading capabilities, we found that SRPL was able to sustain release of its payload over a longer period of time. All release curves were found to be statistically significant from one another (one-way ANOVA with Tukey, *p* < 0.05), with the exception of BSA–(PEG_5000_–Ad)_4_ and BSA–(PEG_5000_–Ad)_8_ in both MP and SRPL pCD. Comparing cumulative release with our total loaded drug values in Figure 6, around 18% of the loaded drug was never released from the pCD MPs while 20% was never released from pCD SRPL.

### 3.7. All mAb–Polymer Conjugates Maintained at Least 70% of Antigen-Recognition Ability

As past literature has noted that mAb–polymer conjugates are at risk of losing antigen specificity after alteration, we utilized a modified ELISA to quantify these changes in structure/function. After modifying the detection mAb of an ELISA sandwich assay, we found that given a constant saturation of ligand (human interleukin-10 (hIL-10), 500 pg/mL), a decrease in overall detection occurred, which we attributed to loss of antigen recognition. Normalizing these values to unmodified positive controls, we were able to obtain a quantitative value for the mAb analog’s ability to recognize its respective antigen (Figure 8). While all groups were statistically different from the positive controls, one-way ANOVA test revealed that the groups were not statistically different from one another (*p* = 0.151).

## 4. Discussion

As pCD has been traditionally used for the prolonged delivery of small-molecule, hydrophobic drugs by complexing with drug ligands in a 1:1 model, this work demonstrates a novel approach for protein-based therapeutic delivery by leveraging the thermodynamic interactions between pCD polymers and protein–polymer conjugates. We successfully synthesized a PEG–Adamantane “tether” group, which was shown to remain unhydrolyzed over the span of 40 days (Figure 3 and Figure 4). We then observed that these high-affinity “tether” groups were able to be successfully conjugated to reduce peptides via maleimide–sulfhydryl chemistry (Figure 5), and that the addition of PEG–Ad groups to peptides significantly increased their affinity for pCD, as shown in decreased K_D_ values between protein–polymer conjugates and *β*-CD (Table 1). This increase in affinity, as correlated in previous studies, yielded significant gains in loading capabilities in pCD polymers (Figure 6) and greatly enhanced drug release kinetics (Figure 7). Compared with unmodified BSA, which was delivered at a limited capacity for only 1 day, our BSA conjugates were able to load in pCD polymers at significantly higher amounts and exhibited a profoundly longer window of delivery (Figure 6 and Figure 7).

It is well known that PEGylation and modification of mAbs and other structure-dependent therapeutics decrease their ability to bind to their respective ligand in vitro and in vivo [42]. We sought to investigate the impact PEG_5000_–Ad conjugation has on mAb antigen recognition in vitro by modifying a sandwich ELISA assay by conjugating the polymer groups to the detection antibody. We found this modified method produced detectable differences in antigen-recognition capabilities between unmodified and modified anti-IL-10 mAbs, and while some mAb function was lost, ~70% specificity was retained between all protein–polymer groups (Figure 8). We recognize this approach to assessing retained receptor recognition is not ideal, and analyzing Fc-mediated binding to Fc gamma receptors or FcRn receptors would provide more accurate insight on retained antibody activity; however, we believe that our simplified, ELISA-derived assay provides an appropriate proxy for antigen recognition for the scope of this study. In addition, due to the low volumes of mAbs we were conjugating, we were not able to complete a molecular weight analysis of the species generated as a result of conjugation. Therefore, future work will need to use a more granular approach to molecular weight analysis, such as mass spectroscopy, to understand the distribution of PEG to antibody ratios.

When measuring affinity, we found that SPR was unable to achieve a statistically significant steady-state curve fitting with chemical species with over two potential complexing domains, namely, BSA–(PEG_5000_–Ad)_4/8_ and mAb–(PEG_5000_–Ad)_4/8_ (Table 1). We hypothesize that because SPR measures “affinity” between a monolayered “host” and mobile “ligand”, it was unable to accurately detect higher-degree complexation. Nonetheless, based on drug loading and drug release kinetics, we can still conclude that an increase in complexation “high-affinity” groups increased the species’ overall affinity for *β*-CD (Figure 6 and Figure 7). Regardless, it is also important to note that K_D_ values decreased nonlinearly (Table 1), suggesting that as more PEG–Ad groups were added, the molecule was unlikely to maximize interactions with pCD (i.e., not all Ad functional groups were complexed with a CD molecule).

Drug loading and delivery results suggest that there may be an upper limit to how many inclusion complexes a single drug payload can form in pCD. Given the similar performance of BSA–(PEG_5000_–Ad)_4_ and BSA–(PEG_5000_–Ad)_8_ in both MP and SRPL pCD (Figure 6 and Figure 7), we hypothesize that despite BSA–(PEG_5000_–Ad)_8′_s average of 5.9 “affinity groups” per molecule, not all of these groups are being utilized to increase thermodynamic interactions, perhaps due to the limited distancing of Ad from the protein itself. Most likely, only two to four groups are actually available for docking in pCD, while the other PEG–Ad groups only add to molecular steric hindrance, which may explain the decreased overall loading efficiency in BSA–(PEG_5000_–Ad)_4/8_ (Figure 6). However, we did observe that the increased “affinity” groups decreased the overall “burst” release that occurred both in MP and SRPL pCD (Figure 7). However, the observation of a “burst” release within the first day suggests that the drug payload was only interacting with the outer surface of the pCD structures and not fully encapsulated within the pCD. This superficial interaction would likely only result in weaker thermodynamic interactions, such as van der Waal forces, and not pCD complexation. Future studies will need to focus on how to maximize the entropy of mixing during the loading phase of the protocol or explore other methods for ensuring that larger payloads, such as peptides and proteins, are properly complexed with the delivery vector; doing so would potentially reduce the “burst” phenomenon observed in Figure 7.

pCD structure appeared to not have a large impact on drug loading capabilities; however, SRPL was observed to have an extended window of delivery of up to 65 days, as opposed to 45–58 days in MP pCD (Figure 7). We predict this difference in delivery stems from topology differences between MP spheres and SRPL “branches”—permeable branching architecture allowed for increased interactions between payload and pCD. Furthermore, we observed a notable increase in optical opacity in SRPL samples after BSA-conjugate loading, which may have been a product of PEG–Ad complexation with pCD.

The idea of “multiplexing” interactions between a chemical species and cyclodextrin inclusion complexes has been consistently shown to enhance loading and delivery of small-molecule therapeutics; however, to the best of our knowledge, this study is the first study that showed that multiplexing thermodynamic interactions can help deliver higher molecular weight payloads (>1 kD) [32,33]. While the exact mechanism of loading and release has not been studied, based on previous cyclodextrin kinematic modeling studies, we suspect that our drug release profiles are the integrated result of thermodynamic interactions, diffusion, and aqueous solubility, which are impacted by our “affinity-group” adamantane hydrolysis from PEG [43,44].

While this study specifically examined protein–polymer conjugates synthesized via maleimide–sulfhydryl chemistry, alternative bioconjugation techniques, such as lysine-based conjugation, can be utilized to generate similar species [45]. For example, the ester bond connecting Ad–PEG can be replaced with an amide for more stable tethers or acid/base reactive tethers for controlled and triggered delivery, respectively. In addition, altering PEG-linker lengths may impact the capacity for pCD binding—our attempt to account for this in our experimental design (choosing PEG_5000_ due to its hydrodynamic radius) seems to have sufficiently distanced our adamantane groups from the protein to promote inclusion in cyclodextrins.

BSA was used as a model drug for this study due to its low cost and ease of detection; however, our outlined synthesis and pCD delivery platform can be applied to extend the drug delivery of any peptide or protein therapeutic with modifiable residues. Especially in situations where local, sustained delivery of protein therapeutics is ideal for positive treatment outcomes, such as the delivery of anti-VEGF antibodies to the retina for treating diabetic retinopathy, the local delivery of chemotherapeutics to a tumor, or the local treatment of rheumatoid arthritis. This platform can be utilized to maintain steady local drug concentration over a period of up to 65 days. We recognize that refillable implants are not currently typical in clinical workflows; however, based on the increasing prevalence of lifelong diseases and chronic conditions, we believe in situ, refillable implants may have a place in clinical practice in the future. Furthermore, while extensive in vivo testing is needed before these protein–polymer conjugates can be utilized for patients, we are encouraged that pCD hydrogels have, historically, performed similarly in in vitro versus in vivo models [24,36,46].

## 5. Conclusions

Herein, we outlined a novel synthesis of protein–polymer conjugates that, when coupled with cyclodextrin delivery platforms, can maintain a sustained release of up to 65 days without largely sacrificing protein structure/function. Antibody and protein-based therapeutics have been increasing in popularity in clinical pharmacology and this study has shown that pCD polymers are suitable for protein drug delivery, not just small-molecule hydrophobic drugs. Compared with traditional administration protocols for peptide therapeutics, many of which involve parenteral injection, pCD hydrogels present a potential effective delivery mechanism. Utilizing a variety of bioconjugation tools, other protein therapeutics can take advantage of pCD’s affinity-based release and can potentially be tuned to fit other relevant clinical applications.

## Figures and Tables

**Figure 1 pharmaceutics-14-01088-f001:**
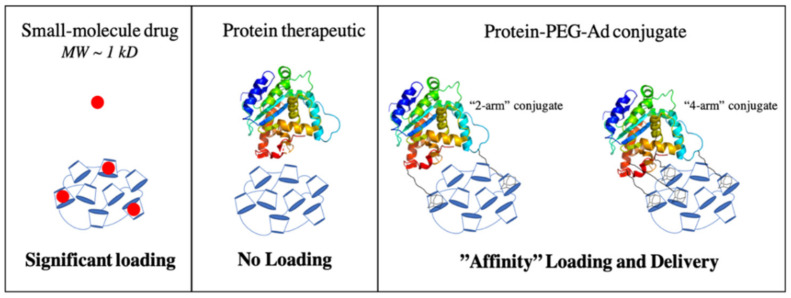
Illustration comparing the loading of small-molecule drugs, protein therapeutics, and protein–PEG–Ad conjugates. Increased conjugation of “PEG-Ad” groups should decrease K_D_, increase thermodynamic interactions between payload and pCD matrix, and ultimately increase loading and prolong release.

**Figure 2 pharmaceutics-14-01088-f002:**
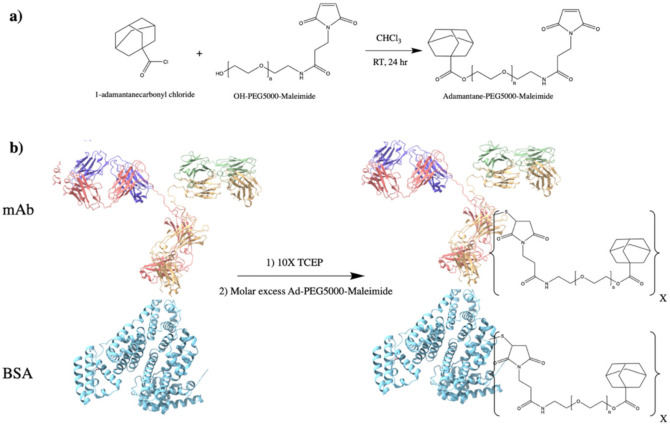
Synthesis overview for protein–PEG–Ad conjugates: (**a**) synthesis of Ad–PEG_5000_–Mal, a nucleophilic addition/elimination reaction; and (**b**) synthesis of protein–(PEG_5000_–Ad)_x_ where X represents the number of PEG_5000_–Ad “tethers” conjugated to each molecule of protein. BSA (PDB entry 4F5S) is used as a model protein, mAb (PDB entry 1IGT) is used as a “functional example” of an antibody therapeutic. Products for mAb synthesis in (**b**) will also produce two fragmented antibodies (Fabs).

**Figure 3 pharmaceutics-14-01088-f003:**
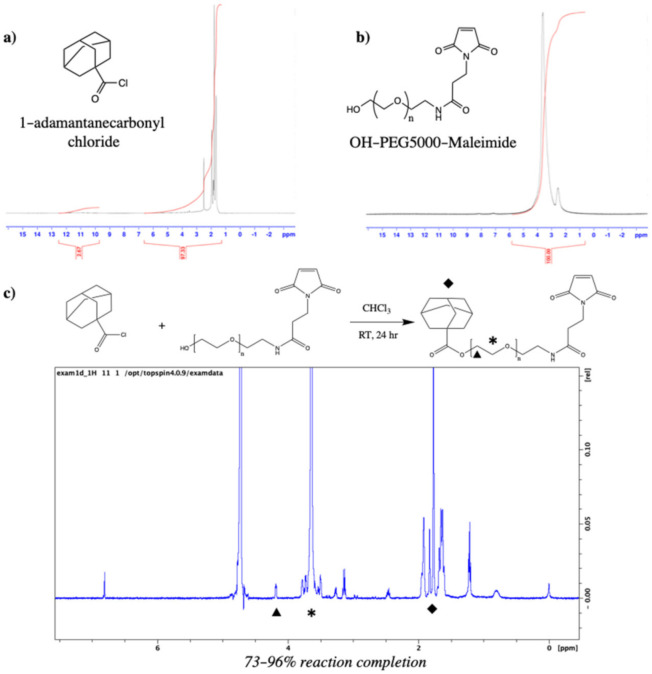
(**a**) ^1^H NMR (DMSO-*d_6_*) of 1-adamantanecarbonyl chloride which confirmed a 97.33% purity; (**b**) ^1^H NMR (DMSO-*d_6_*) of maleimide–PEG_5000_–hydroxyl; and (**c**) Ad–PEG_5000_–Mal ^1^H NMR (DMSO-*d_6_*), with unique peaks at δ = 1.7–1.9 ppm (Ad hydrocarbons, ◆), δ = 4.2 ppm (terminal −CH_2_− of PEG, ▴), and δ = 3.6 ppm (−CH_2_− PEG repeat units, *).

**Figure 4 pharmaceutics-14-01088-f004:**
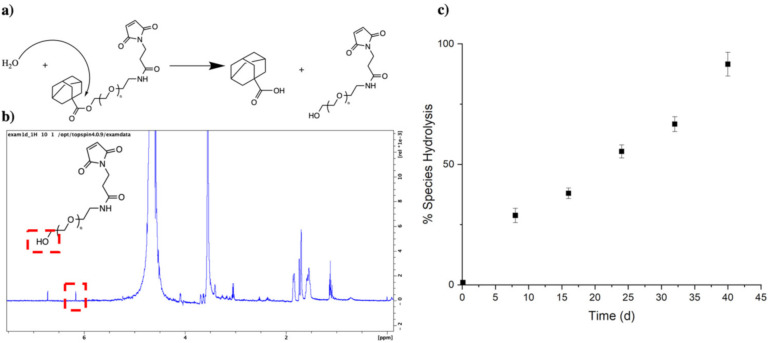
(**a**) Hydrolysis mechanism for PEG–Ad; (**b**) ^1^H NMR (D_2_O) with Bruker 300 MHz, noting a hydrolysis peak at δ = 6.1 ppm and a reference peak at δ = 4.2 ppm (−CH_2_−); and (**c**) a species percentage was quantitively obtained from hydrolysis peak δ = 6.1 ppm.

**Figure 5 pharmaceutics-14-01088-f005:**
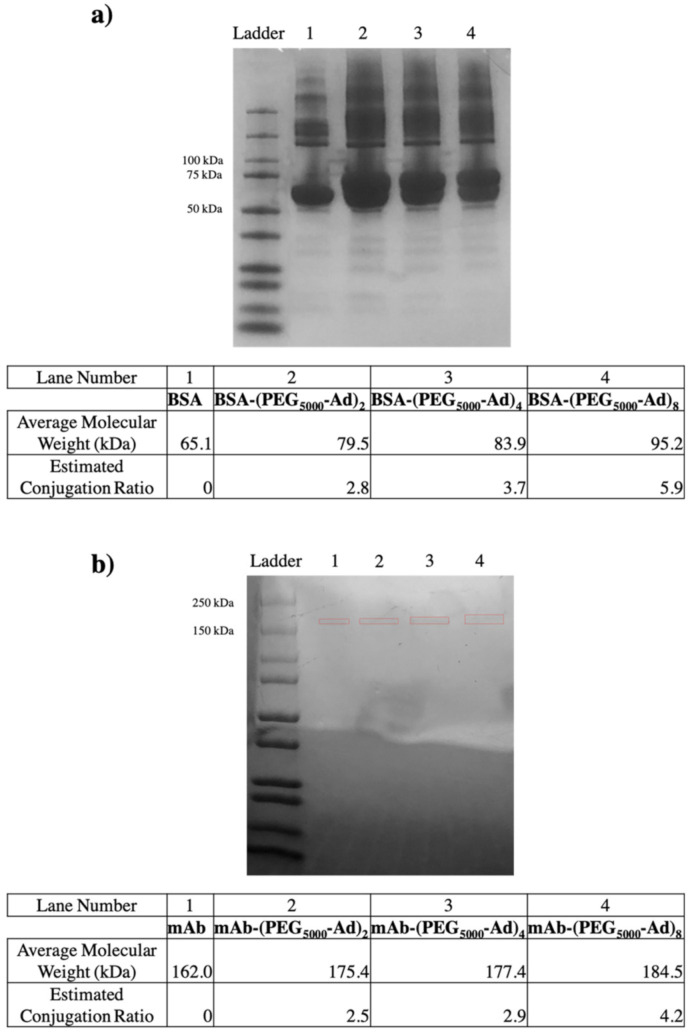
Nonreducing SDS-PAGE gel electrophoresis (12–20% Tris Glycine) of (**a**) BSA conjugates found between 79–95 kDa and (**b**) mAb conjugates found between 175–185 kDa. Average molecular weights were obtained based on ladder band locations in ImageJ. mAb bands were boxed in red to increase visibility, as loaded protein concentration was low.

**Figure 6 pharmaceutics-14-01088-f006:**
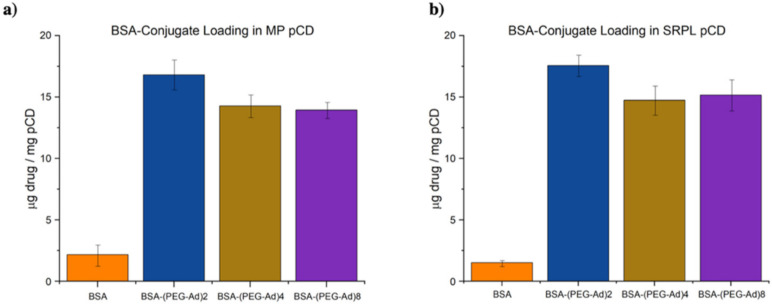
Total loaded drug normalized to pCD weight (10 mg) for BSA–polymer conjugates for (**a**) pCD MPs and (**b**) pCD SRPL.

**Figure 7 pharmaceutics-14-01088-f007:**
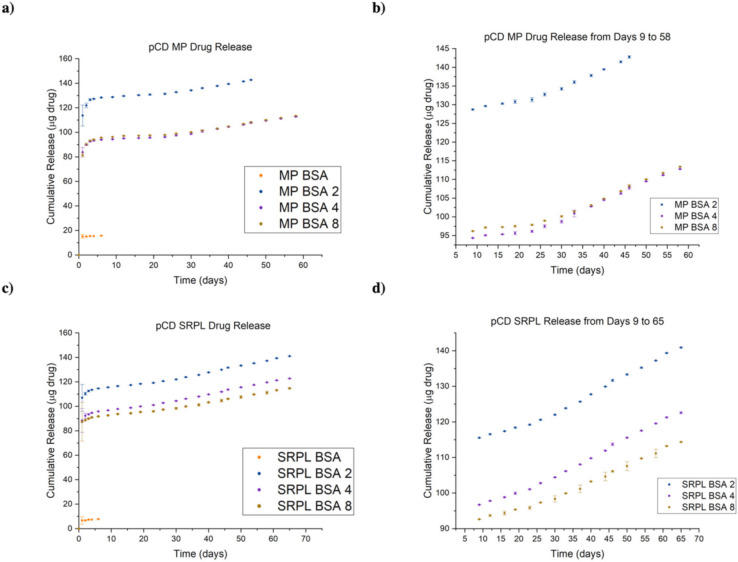
Cumulative drug release curves for BSA–polymer conjugates from (**a**) pCD MPs, (**b**) pCD MPs zoomed in on release days 9–65, and (**c**) pCD SRPL and (**d**) pCD SRPL zoomed in on release days 9–65. All datasets were found to be statistically significant from one another (one-way ANOVA with Tukey, *p* < 0.05), except BSA 4 and BSA 8 curves in both MP and SRPL trials.

**Figure 8 pharmaceutics-14-01088-f008:**
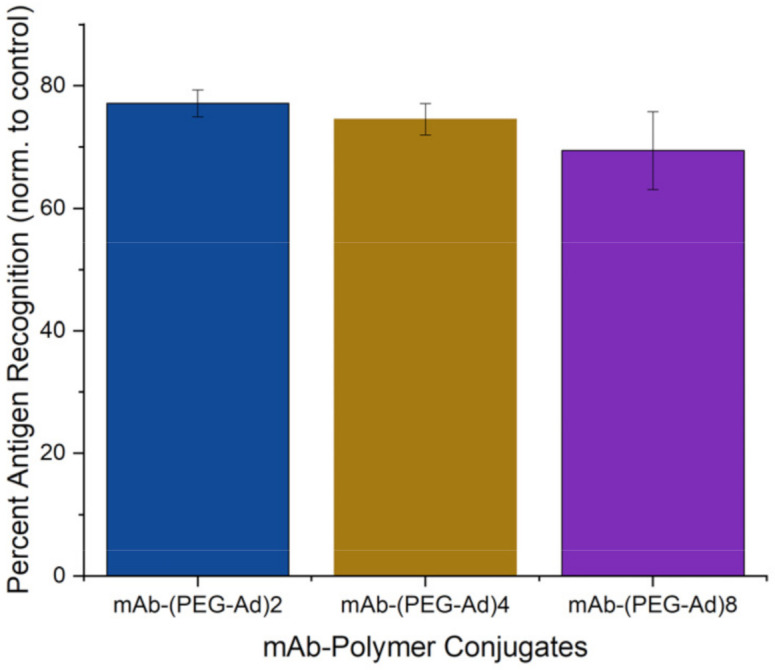
Quantitative estimation of mAb (anti-IL10) structure/function after PEG_5000_–Ad conjugation. Positive controls (according to MabTag protocol, using unmodified primary antibodies) were assumed to have 100% antigen recognition.

**Table 1 pharmaceutics-14-01088-t001:** SPR kinetics results for unmodified and modified protein–polymer conjugates against *β*-CD immobilized CM5 chip, diH_2_O running buffer.

Interaction (Host/Ligand)	K_D_ (M)	Binding Free Energy (KJ/mol)
*β*CD/BSA	0.001293 ^1^	−16.5
*β*CD/BSA–(PEG_5000_–Ad)_2_	7.966 × 10^−5 1^	−23.4
*β*CD/BSA–(PEG_5000_–Ad)_4_	9.090 × 10^−9^	−45.9
*β*CD/BSA–(PEG_5000_–Ad)_8_	4.099 × 10^−10^	−53.6
*β*CD/mAb	0.0001443 ^1^	−21.9
*β*CD/mAb–(PEG_5000_–Ad)_2_	8.755 × 10^−10 1^	−52.9
*β*CD/mAb–(PEG_5000_–Ad)_4_	3.012 × 10^−11^	−54.1
*β*CD/mAb–(PEG_5000_–Ad)_8_	4.515 × 10^−13^	−70.4

^1^ K_D_ values were within the model confidence interval with Chi^2^ values below 10% of the maximum SPR response.

## Data Availability

Not applicable.

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
