# Peer review of "Leveraging Affinity Interactions to Prolong Drug Delivery of Protein Therapeutics"

_pharmaceutics, 2022, doi:10.3390/pharmaceutics14051088_

Round 1

Reviewer 1 Report

The article “Leveraging Affinity Interactions to Prolong Drug Delivery of Protein Therapeutics” (Manuscript Number: pharmaceutics-1715810) proposed the novel synthesis of protein-polymer conjugates that are capped with a “high affinity” adamantane. This study has shown that pCD polymers are suitable for protein drug delivery, not just small olecule hydrophobic drugs. Utilizing a variety of bioconjugation tools, other protein therapeutics can take advantage of pCD’s affinity-based release and can potentially be tuned to fit other relevant clinical applications. In general, the experiment is well-designed and the conclusion is supported by the experiment and results. However, there are still some issues to be addressed. It needs revisions and improvements before the acceptance:

  1. In the first paragraph of the introduction, why did the author switch from protein therapy to m onoclonal antibody therapy but not continue to explain this method?
  2. When mentioned affinity-based drug delivery systems, the background of polymerized cyclodextrin should be expanded. The following articles are relevant to your work which are suggested to be read: 10.1039/C9CS00839J; 10.1016/j.jobab.2021.04.002.
  3. The manuscript still lacks an adequate use of the published literature to introduce the background of this study. For example, in the section of the introduction, “While many groups have investigated pCD to deliver small molecules like Rifampicin and other smaller therapeutics like RNA…”. Please give some specific research to illustrate this statement. The following article is helpful to your work which is suggested to be read:1038/s41565-021-00976-3.
  4. In the manuscript, why did the author choose adamantane to improve the affinity of the pCD polymers?
  5. In the section of Materials and Methods, there are also some inconsistencies in format, for example “OC”. Please check the full manuscript and modify it. The following articles is helpful to your work which is suggested to be read: 10.1016/j.jobab.2021.01.002.
  6. The quality of Figure 3 and Figure 4 is poor. Please improve the clarity of the pictures.
  7. In the manuscript, it is suggested that the author can explain the pictures separately to make the explanation more detailed and clarify the meaning of the pictures.
  8. In Figure 8, What is the content of the positive control regimen and the different regimen tested separately? It is suggested that the author increase the initial data for comparison before normalization.
  9. In line 420, “we suspect that our drug release profiles are the integrated result of thermodynamic interactions, diffusion, and aqueous solubility….”, Please list some observations in the literature that would support these assumptions.

Author Response

Thank you for your revisions - please see the attachment.

Reviewer 2 Report

Comments to the Authors:

This manuscript describes two series of functional hydrogels designed and synthesized by polymerized cyclodextrin and two proteins (bovine serum albumin and monoclonal antibodies) that are covalently link by multiple adamantanes. The release efficiency of hydrosol was tested, and the recognition ability of antigen was at least 70%. The protein-hydrogel system is an intentional system in the protein drugs. This manuscript can be published in Pharmaceutics after the following revisions.

  1. Page 1, line 27, “autoimmunity/inflammatory diseases” and line 31 “autoimmune diseases”. It should be an adjective that modifies a noun.
  2. Page 5, line 152, “DMSO-D6 or Chloroform-D solvent” should be changed to “DMSO-d6 or Chloroform-d solvent”. That means “D” is in italics.
  3. Page 6, line 170, “β-cyclodextrin” should be changed to “β-cyclodextrin”. That means “β” is in italics.
  4. Line 181, “81.88 ± 36.86 um” should be changed to “81.88 ± 36.86 μm”.
  5. Line 184, “at 70℃” should be change to “at 70 ℃”.
  6. Line 186, “45 uL” should be change to “45 μL”.
  7. Line 210, “100 ug/ml” should be change to “100 μg/mL”.
  8. Page 7, line 235, “n=3” should be change to “n = 3”.
  9. Page 8, line 266, “1H NMR (DMSO-d6)” should be change to “1H NMR (DMSO-d6)”. That means “d” is in italics.
  10. The picture “Figure6” at the page 10, line 317, with “Comparing cumulative release with our total release values in Figure 6,” is inconsistent with the text and the author should check it carefully.
  11. The uppercase and lowercase letters of title words in the references should be unified in the references “4. 5. 7. 12. 13. 14. 16. 20. 21. 23. 27. 31.” Author should complete the volume and page numbers in the references “12. 24. 26. 30.”
  12. Whether hydrogel dissolves in water, if not, how to the apply in living organisms. The reaction of hydrolysis in an organism (as in the blood), because it is measured in the DMSO.
  13. Page 2, Introduction, the authors said “Broadly, “affinity-based” delivery encompasses any matrix/guest system that leverages receptor/ligand, electrostatic/ionic, hydrogen bonding, hydrophobic or van der Waal interactions to increase the thermodynamic interactions between the matrix host and the guest payload.”. Some drug delivery systems based on host and guest should be mentioned, the following recently published important related paper should be cited:  Chem. Soc. Rev. 2021, 50, 2839; Adv Mater. 2022, 34, 2106388.

Author Response

We thank the reviewer for their suggestions - please see the attachment

Reviewer 3 Report

This manuscript describes a new synthesis of protein-polymer conjugates that are capped with a “high affinity” adamantane. Nowadays antibody and protein-based therapeutics have been increasing in popularity in clinical pharmacology. In this paper, the authors have shown that cyclodextrin-based polymers are suitable for protein drug delivery, not just small-molecule hydrophobic drugs. The authors report encouraging data about protein-polymer conjugates that, when coupled with cyclodextrin delivery platforms, can maintain a sustained release of up to 65 days without largely sacrificing protein structure/function.

The manuscript provides a good amount of data and it is relevant to the journal. I recommend this manuscript for publication in Pharmaceutics.

Author Response

We thank the reviewer for their approval - we hope any supplemental edits made from other reviewer suggestions will further improve the quality of the study.